# Factors associated with mortality of elderly people due to COVID-19: Protocol for systematic review and meta-analysis

**Danyllo do Nascimento Silva Junior**[1,2], **Ádala Nayana de Sousa Mata**[2,3,4], **Gidyenne Christine Bandeira Silva de Medeiros**[1,2,5], **Marilane Vilela Marques**[1,2], **Thais Teixeira dos Santos**[1,2], **Maria Eduarda de Sousa Monteiro**[2,4], **Gabriela Góis Costa**[2,4], **Eleonora d´Orsi**[6], **Eva Vegue Parra**[7], **Grasiela Piuvezam**[1,2,8] *

1 Postgraduate Program of Public Health, Federal University of Rio Grande do Norte (UFRN), Natal, RN, Brazil, 2 Systematic Review and Meta-Analysis Laboratory (Lab-Sys/CNPq), UFRN, Natal, RN, Brazil, 3 Postgraduate Program in Education, Work and Innovation in Medicine, Federal University of Rio Grande do Norte (PPGETIM), UFRN, Caicó, RN, Brazil, 4 Multicampi School of Medical Sciences of Rio Grande do Norte, Federal University of Rio Grande do Norte (UFRN), Caicó, RN, Brazil, 5 Department of Nutrition, Federal University of Rio Grande do Norte (UFRN), Natal, RN, Brazil, 6 Department of Public Health, Federal University of Santa Catarina (UFSC), Florianópolis, SC, Brazil, 7 Universidad Católica San Antonio de Murcia (UCAM), Murcia, Spain, 8 Department of Public Health, Federal University of Rio Grande do Norte (UFRN), Natal, RN, Brazil

* gpiuvezam@yahoo.com.br

**Data Availability Statement:** No datasets were generated or analysed during the current study. All relevant data from this study will be made available upon study completion.

## Abstract

### Introduction

The COVID-19 pandemic has become a significant health crisis, marked by high mortality rates on a global scale, with mortality from the disease being notably concentrated among the elderly due to various factors.

### Objective

This study aims to investigate the biological and non-biological factors associated with COVID-19 mortality rates among the elderly worldwide.

### Methods

The following databases will be consulted: PubMed, Scopus, EMBASE, Web of Science and ScienceDirect. Longitudinal observational studies (cohort and case-control—risk factors) will be included. The risk of bias, defined as low, moderate, high, will be assessed using the National Heart, Lung and Blood Institute (NHLBI) Quality Assessment Tool for observational cohort and cross-sectional studies. Two independent authors will conduct the searches, and any possible disagreements will be resolved by a third author. Heterogeneity between study results will be assessed using a standard $X^2$ test with a significance level of 0.05, and an $I^2$ value will be calculated to further assess heterogeneity. The random effects model for meta-analyses will be adopted to distribute the weight between the studies and standardize their contributions. The meta-analyses will be conducted using RevMan software.

**Funding:** This study was financed in part by the Coordenação de Aperfeiçoamento de Pessoal de Nível Superior - Brasil (CAPES) - Finance Code 001. The funders did not and will not have a role in study design, data collection and analysis, decision to publish, or preparation of the manuscript.

**Competing interests:** The authors have declared that no competing interests exist.

## Discussion

Despite the numerous publications on COVID-19 mortality among the elderly, there is still a gap in knowledge, as there is no systematic review and meta-analysis that summarizes the main biological and non-biological associated factors globally.

## Conclusion

The results of this study will consolidate the latest evidence and address gaps in the overall understanding of biological or non-biological associated factors. This knowledge will facilitate the development of appropriate health strategies for this demographic group and pave the way for further research.

## Trial registration

PROSPERO (CRD42023400873).

## Introduction

COVID-19 infection affects people indiscriminately, making it a global health crisis with significant mortality rates. However, despite its indiscriminate nature, the elderly population has experienced concentrated mortality since the onset of the pandemic. This demographic, for various reasons, is more vulnerable to complications and fatalities caused by the SARS-CoV-2 virus [1, 2].

The risk of SARS-CoV-2 infection and its clinical progression were initially challenging to predict, and the factors that contribute to increased susceptibility and a more severe course of the disease are currently under ongoing research, with full understanding not yet achieved. Broadly categorized as an infectious disease, primarily affecting the respiratory system, COVID-19 exhibits diverse clinical implications, spanning from mild to severe manifestations [1–3].

Longevity, once a global success story for societies and public health policies, has brought forth new challenges, particularly in the context of global health crises. The physiological aspects of aging contribute to the vulnerability of the elderly population, impacting the effectiveness of the immune system and increasing susceptibility to morbidity and mortality from infectious diseases [3]. Moreover, numerous studies indicate that various factors influence the progression and outcome of COVID-19 in the elderly, differing from other age groups [1, 4, 5].

Specifically, studies in various countries and regions have identified advanced age, male gender, and the presence of comorbidities as the primary biological factors associated with COVID-19 mortality in the elderly [1, 6]. Common comorbidities such as hypertension, diabetes, cardiovascular and respiratory diseases, and dementia are strongly linked to increased severity and subsequent mortality from COVID-19 [3, 6, 7].

To curb the pandemic, vaccines have been developed, with the initial doses administered in December 2020 [8]. Widespread vaccination has proven effective in reducing morbidity and mortality from COVID-19, extending beyond direct protection to create indirect shielding for the entire community, thereby lowering the risk of infection for vulnerable individuals [8, 9].

On the non-biological front, various factors, particularly behavioral ones (such as mask usage, social distancing/isolation, and hand hygiene), as well as sociodemographic factors,

strongly correlate with the spread of COVID-19 and the mortality of the elderly population worldwide [5, 10, 11].

In this study, we intend to investigate both biological and non-biological factors affecting the mortality of the elderly population from COVID-19. The biological factors under scrutiny include age categories (60–69; 70–79; 80+), gender (female/male), comorbidities (such as hypertension, diabetes, respiratory, and cardiovascular diseases), and vaccination status (whether vaccinated or not, and the number of doses administered). Additionally, non-biological factors to be examined encompass behavioral aspects (such as the use of masks, adherence to social distancing/isolation, and hand hygiene) and sociodemographic factors (including income, education, and whether individuals live alone or not).

Deepening our understanding of these factors holds the potential to generate crucial insights into the mortality patterns observed in the elderly population affected by COVID-19. Despite the wealth of studies produced and published [3, 12, 13], a comprehensive worldwide systematic analysis specifically focused on elderly individuals is currently lacking. Addressing this gap is essential to fortify our understanding of the unique challenges faced by the elderly, enabling more robust health planning. This, in turn, facilitates the development of differentiated approaches and appropriate interventions tailored to the specific health characteristics of this demographic, particularly in the context of health crises like the COVID-19 pandemic.

In view of the foregoing, the aim of this study is to investigate the biological and non-biological factors associated with mortality rates in the elderly population due to COVID-19.

## Materials and methods

### Study registration

This protocol was registered in the International Prospective Register of Systematic Reviews (PROSPERO) on 01 March 2023 (CRD42023400873), according to Preferred Reporting Items for Systematic Reviews and Meta-Analyses Protocols (PRISMA) [14] S1 and S2 Files.

Secondary data extracted from the scientific literature will be used, which is why there will be no need for prior ethical approval.

### Eligibility criteria

**Inclusion criteria.** Articles from peer-reviewed journals that meet eligibility criteria based on study population, exposure, outcome, and types of studies (PICOS) will be included in this review.

Inclusion criteria will be: (a) population–individuals aged 60 or over; (b) exposure–testing positive for SARS-CoV-2; (c) results–factors associated with the mortality rate of the elderly population from COVID-19; (d) types of studies–longitudinal observational (cohort and case-control–risk factors).

**Exclusion criteria.** For exclusion, the following criteria were defined: (a) population–institutionalized individuals and/or those with dementia and/or death data under the age of 60; (b) exposure–studies on pandemic deaths from non-COVID-19 causes; (c) results–studies that do not present the factors associated with deaths of elderly people from COVID-19; (d) types of studies–publications that do not answer the guiding question of this systematic review. Cross-sectional observational studies (prevalence and ecological), other systematic reviews, scoping reviews, thesis, and dissertations will also be excluded.

There will be no language restrictions. There will be a restriction regarding publication time (2020–2023) due to the temporality of the COVID-19 pandemic.

## Search strategies and study identification

Initially, eligible studies will be identified by the criteria established from the search strategies with the keywords indexed in the Medical Subject Headings (MeSH), a combination of descriptors related to factors associated with the mortality of elderly people due to COVID-19 in the world. The search equations will be accompanied by the Boolean operators OR and AND S3 File. Next, the following electronic databases will be used for searches: PubMed, Scopus, EMBASE, Web of Science and ScienceDirect. The searches in these databases will be carried out from the access data of the researchers provided by the Federal University of Rio Grande do Norte (UFRN), which gives open access to all articles in each database through the "Capes' Journals Portal" in Brazil.

## Data extraction

After checking and eliminating potential duplicates, all articles resulting from database searches using the specified equations will undergo independent review of titles and abstracts by two reviewers. Through this process, studies will be selected based on the defined eligibility criteria. The Rayyan QCRI program will be employed for the systematic review's study selection [15].

During the selection of studies for the systematic review, the references of the studies included in the full-text evaluation phase will be scrutinized to identify potentially relevant studies that were not considered in earlier phases. In instances of conflict or disagreement between reviewers at any stage, a third researcher will be consulted for resolution. If data is missing or unclear, attempts will be made to contact the study authors for clarification.

The data to be extracted from the selected studies will include (1) author and year of publication, (2) name of the journal, (3) study design, (4) country of origin of the study, (5) objective, (6) sample size, (7) period of data collection, (8) statistical test used, (9) biological factors: age (60–69; 70–79; 80+), gender (female/male), comorbidities (hypertension, diabetes, respiratory and cardiovascular diseases) and vaccination (vaccinated or not and the number of doses), (10) non-biological factors: behavioral (use of masks, social distancing/isolation, hand hygiene) and sociodemographic (income, education, living alone or not), (11) COVID-19 variants studied, (12) funding source, (13) authors' conclusions. A pre-designed and previously tested spreadsheet (Microsoft Excel) will be used to record the data.

## Risk of bias and evaluation by the grading of recommendations, assessment, development and evaluation (GRADE)

Reviewers will assess the risk of bias in selected studies using the NHLBI's Quality Assessment Tool for Observational Cohort and Cross-Sectional Studies and Quality Assessment of Case-Control Studies (https://www.nhlbi.nih.gov/health-topics/study-quality-assessment-tools). The risk of bias will be ranked using predetermined criteria as follows: low, moderate, high.

To evaluate the quality of evidence in the studies included, the Grading of Recommendations, Assessment, Development, and Evaluation (GRADE) will be utilized [16]. In the event of uncertainties or discrepancies in the assessments, a third researcher will be consulted. We will make an effort to retrieve any missing data by reaching out to the corresponding author or co-author via email, phone, or mail to request the necessary information. If contact cannot be established, the data will be excluded from our analysis, and this will be addressed in the discussion section. The funnel plot will be used to assess publication bias. In addition, sensitivity analysis could be considered to give greater reliability to the results of the review [17].

## Data synthesis and analysis

Data will be presented in tables and charts and in a narrative way to describe the characteristics of the included studies. A meta-analysis will be conducted if the studies are sufficiently homogeneous. Heterogeneity between study results will be evaluated using a standard $X^2$ test with a significance level of 0.05. An assessment of heterogeneity with $I^2$ value will be performed. Heterogeneity of around 25% will be considered low, around 50%, moderate, and around 75%, high. The random effects model for meta-analyses will be adopted to distribute the weight among the studies and standardize their contributions. Meta-analyses will also be performed using RevMan software (version 0.1.0). In addition, meta-regression could facilitate the analysis of confounding factors related to mortality in the elderly, enabling the identification of those truly associated with COVID-19.

Descriptive analysis will be performed using SPSS Statistics 28. Weighted means and 95% CI will be calculated for continuous variables. We will calculate the hazard ratio and 95% CI for each dichotomous data outcome. The Grading of recommendations, assessment, development, and evaluation (GRADE) approach will guide the assessment regarding the overall confidence of each selected study, with a view to rating the quality of the evidence and the strength of the recommendations.

## Ethics and dissemination

Ethical approval is not required for this protocol, as it pertains to a systematic review. In this study, participants are not actively recruited, and data are not collected directly from them. The findings of the review will be disseminated through peer-reviewed publications.

## Discussion

There is a substantial body of literature worldwide focused on the mortality of elderly individuals due to COVID-19. However, as of yet, no systematic review comprehensively summarizes the primary factors associated with COVID-19 mortality in the global elderly population, considering diverse countries across all continents and the various contributing factors.

A systematic review and meta-analysis investigating mortality factors among elderly Italians diagnosed with coronavirus, residing in institutions or hospitalized due to the disease, identified dementia, diabetes, chronic kidney disease, and hypertension as the primary diseases linked to mortality in this population. The authors hypothesized that this association is attributed to the high prevalence of these diseases among the elderly [3]. However, the study's scope was limited as it did not explore COVID-19-related deaths outside these environments.

Findings from another systematic review, encompassing a large dataset from multiple studies, consistently highlighted comorbidities, gender, age, smoking, obesity, acute kidney injury, and D-dimer as clinical risk factors for fatal outcomes associated with the coronavirus [18]. Similarly, a comprehensive systematic review of 207 studies identified 49 variables offering valuable prognostic information about mortality and/or severe illness in COVID-19 patients [19]. Notably, both studies incorporated data from patients of all ages, failing to address the specificities of the elderly population.

The other study identified advanced age, male gender, dyspnea, and dementia as factors associated with a higher risk of death from COVID-19 in the elderly population [12]. However, these studies relied on a limited number of databases, restricting the global applicability of their results.

A notable limitation in the existing systematic review studies on COVID-19 mortality among the elderly pertains to a narrow focus on specific factors, neglecting a comprehensive examination of all associated factors. For instance, one study highlighted that elderly

individuals with dementia diagnosed with COVID-19 face a higher risk of mortality compared to those without dementia [20]. Another study found that, overall, frailty among older adults was linked to higher rates of COVID-19-related mortality compared with non-frail counterparts [21]. A third study concluded that comorbidities contribute to increased COVID-19 mortality among the elderly but relied on a single database [13].

In this context, there is a notable absence of systematic review studies consolidating knowledge about both biological and non-biological factors among elderly individuals who succumbed to SARS-CoV-2 infection, considering global data. Given the consensus that the elderly population is the most vulnerable demographic to serious outcomes and deaths during pandemics, addressing these gaps is crucial for the informed development of public policies, enabling countries to minimize the impacts on this population, particularly during health crises such as the COVID-19 pandemic.

## Strengths, limitations and implications

This study has some limitations. There may be significant heterogeneity in the data from the studies, which could limit the meta-analysis. With regard to the exclusion criterion of cross-sectional studies, we recognize that these studies collect data at a single point in time and therefore have limitations in terms of causal inference and temporal follow-up. However, we recognize that the inclusion of several study designs could provide a more comprehensive understanding of the subject.

The study has also significant potential. Our review aims to address some of the methodological limitations identified in previous systematic reviews on this subject, including restrictions related to research time, a restricted number of databases, investigation limited to local data (one or a few countries) and research focused only on biological aspects. In this sense, we believe that our protocol proposes an unprecedented revision by broadening mortality research to include clinical (biological) and behavioral (non-biological) aspects. This approach imposes no restrictions on location or language and extends over an extended period (2020–2023), incorporating classic epidemiological designs and research in a larger number of databases. In addition, our inclusion criteria involve studies that used gold standard mechanisms (such as RT-PCR) to confirm COVID-19, increasing the credibility of the included studies.

Regarding the implications, the results of this comprehensive global study can contribute to the planning and implementation of interventions targeting the elderly to reduce mortality due to COVID-19. In addition, it offers an opportunity for different countries to mitigate the impacts of pandemics/health crises on the elderly population, which is generally more vulnerable than other age groups.

## Conclusions

In this context, there is a notable absence of systematic review studies consolidating knowledge about both biological and non-biological factors among elderly individuals who succumbed to SARS-CoV-2 infection, considering global data. Given the consensus that the elderly population is the most vulnerable demographic to serious outcomes and deaths during pandemics, addressing these gaps is crucial for the informed development of public policies, enabling countries to minimize the impacts on this population, particularly during health crises such as the COVID-19 pandemic.

## Supporting information

**S1 Table. List of abbreviations used in the text.**
(DOCX)

**S1 File. PRISMA study selection flowchart.** *From*: Page MJ, McKenzie JE, Bossuyt PM, Boutron I, Hoffmann TC, Mulrow CD, et al. The PRISMA 2020 statement: an updated guideline for reporting systematic reviews. BMJ 2021;372: n71. doi: 10.1136/bmj.n71.
(DOCX)

**S2 File. PRISMA 2020 checklist.**
(DOC)

**S3 File. Search equation to be used in each database.** Own authorship.
(DOCX)

## Author Contributions

**Conceptualization:** Danyllo do Nascimento Silva Junior, Ádala Nayana de Sousa Mata, Gidyenne Christine Bandeira Silva de Medeiros, Marilane Vilela Marques, Thais Teixeira dos Santos, Gabriela Góis Costa, Eleonora d´Orsi, Eva Vegue Parra, Grasiela Piuvezam.

**Data curation:** Danyllo do Nascimento Silva Junior, Ádala Nayana de Sousa Mata, Gidyenne Christine Bandeira Silva de Medeiros, Eleonora d´Orsi, Eva Vegue Parra, Grasiela Piuvezam.

**Formal analysis:** Danyllo do Nascimento Silva Junior, Ádala Nayana de Sousa Mata, Gidyenne Christine Bandeira Silva de Medeiros, Marilane Vilela Marques, Thais Teixeira dos Santos, Maria Eduarda de Sousa Monteiro, Gabriela Góis Costa, Eleonora d´Orsi, Eva Vegue Parra, Grasiela Piuvezam.

**Investigation:** Danyllo do Nascimento Silva Junior, Marilane Vilela Marques, Thais Teixeira dos Santos, Maria Eduarda de Sousa Monteiro, Gabriela Góis Costa, Grasiela Piuvezam.

**Methodology:** Danyllo do Nascimento Silva Junior, Ádala Nayana de Sousa Mata, Gidyenne Christine Bandeira Silva de Medeiros, Marilane Vilela Marques, Thais Teixeira dos Santos, Eleonora d´Orsi, Eva Vegue Parra, Grasiela Piuvezam.

**Supervision:** Grasiela Piuvezam.

**Writing – original draft:** Danyllo do Nascimento Silva Junior.

**Writing – review & editing:** Danyllo do Nascimento Silva Junior, Ádala Nayana de Sousa Mata, Gidyenne Christine Bandeira Silva de Medeiros, Thais Teixeira dos Santos, Grasiela Piuvezam.

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
