## [Decision Letter · Decision Letter 0]

8 Oct 2023

PONE-D-23-20685Factors associated with mortality of elderly people due to Covid-19: Protocol for systematic review and meta-analysisPLOS ONE

Dear Dr. Piuvezam,

Thank you for submitting your manuscript to PLOS ONE. After careful consideration, we feel that it has merit but does not fully meet PLOS ONE’s publication criteria as it currently stands. Therefore, we invite you to submit a revised version of the manuscript that addresses the points raised during the review process.

We look forward to receiving your revised manuscript.

Kind regards,

Rocco Franco

Academic Editor

PLOS ONE

Additional Editor Comments:

Dear Authors,

I suggest making the changes requested by the reviewers.

Regards

Reviewers' comments:

Reviewer's Responses to Questions

**Comments to the Author**

1. Does the manuscript provide a valid rationale for the proposed study, with clearly identified and justified research questions?

Reviewer #1: Yes

Reviewer #2: Yes

Reviewer #3: Yes

2. Is the protocol technically sound and planned in a manner that will lead to a meaningful outcome and allow testing the stated hypotheses?

Reviewer #1: Yes

Reviewer #2: Yes

Reviewer #3: Partly

3. Is the methodology feasible and described in sufficient detail to allow the work to be replicable?

Reviewer #1: Yes

Reviewer #2: Yes

Reviewer #3: Yes

4. Have the authors described where all data underlying the findings will be made available when the study is complete?

Reviewer #1: Yes

Reviewer #2: Yes

Reviewer #3: Yes

5. Is the manuscript presented in an intelligible fashion and written in standard English?

Reviewer #1: Yes

Reviewer #2: Yes

Reviewer #3: No

6. Review Comments to the Author

You may also provide optional suggestions and comments to authors that they might find helpful in planning their study.

Reviewer #1: Dears,

The paper has well-designed research methods, appropriate statistical analysis and a relatively good interpretation of the results.

-Please be sure to use only keywords accordingly to medical subject headings (Mesh word) for a better indexing.

I suggest you add a table with the list of abbreviations used in the text.

I suggest you implement the abstract in order to make it more understandable to authors.

The introduction should be expanded perhaps by adding a section on temporomandibular disorders. I recommend some references:[10.3390/jcm12072652];[10.1111/joor.13496]

The conclusion is in accordance with the objectives of the research, its results and their interpretation, as well as the relevant literature.

Regards

Reviewer #2: The present work is very thorough and detailed.

I would like to ask the author why they removed the hospitalized elderly in the exclusion criteria, considering that most covid deaths occurred in hospital facilities. Modify the conclusions to strengthen the sense of the review

Reviewer #3: The authors present a systematic review protocol with the objective of evaluating factors associated with higher mortality from Covid-19 in the elderly.

Although the authors state in the introduction "there is still no systematic and comprehensive evidence worldwide on such issues specifically with elderly people" (lines 113 and 114), a quick search finds systematic reviews with meta-analysis with the same objective already published (doi: 10.1177/23337214211057392./doi: 10.3390/ijerph18158008/ https://doi.org/10.1556/2060.2022.00206) and the authors need to point out in the protocol whether what they propose may indicate different results from those published.

Finally, the protocol needs to consider the possibility of meta-regression in the methodology as it is a systematic review of observational studies, may have a shorter and more objective introduction and needs to be reviewed by a native English speaker to correct grammatical errors. As a result, there is no quality or originality for the protocol to be published.

7. PLOS authors have the option to publish the peer review history of their article (what does this mean?). If published, this will include your full peer review and any attached files.

Reviewer #1: No

Reviewer #2: No

Reviewer #3: **Yes: **Ricardo Ney Cobucci

---

## [Author Response · Author response to Decision Letter 0]

24 Nov 2023

Dear reviewers, below we send you our thoughts on your comments. Thank you in advance for your cooperation.

2. Is the protocol technically sound and planned in a manner that will lead to a meaningful outcome and allow testing the stated hypotheses?

Reviewer #1: Yes

Reviewer #2: Yes

Reviewer #3: Partly

Thank you for taking the time to review our manuscript and for your comments. Considering the relevance of summarizing the current global evidence on biological and non-biological factors associated with the mortality rate of the elderly due to Covid-19, we conducted our research questions according to the dimensions explored in the construction of the review. The research questions, in addition to being included in the registered PRISMA-P protocol, are fully described in the methodology of the manuscript, and were constructed in accordance with the justification presented in the Introduction section.

5. Is the manuscript presented in an intelligible fashion and written in standard English?

Reviewer #1: Yes

Reviewer #2: Yes

Reviewer #3: No

We would like to inform you that we have made some adjustments to improve the fluidity and clarity of the English text.

6. Review Comments to the Author

Reviewer #1: Dears,

The paper has well-designed research methods, appropriate statistical analysis and a relatively good interpretation of the results.

-Please be sure to use only keywords accordingly to medical subject headings (Mesh word) for a better indexing.

I suggest you add a table with the list of abbreviations used in the text.

I suggest you implement the abstract in order to make it more understandable to authors.

The introduction should be expanded perhaps by adding a section on temporomandibular disorders. I recommend some references:[10.3390/jcm12072652];[10.1111/joor.13496]

The conclusion is in accordance with the objectives of the research, its results and their interpretation, as well as the relevant literature.

All the terms entered in the search key have been reviewed and are in MESH as descriptors or between terms.

The table with the list of abbreviations was inserted in the article.

We understand that temporomandibular disorders are not part of the scope of this review, as an outcome or criterion to be investigated. Therefore, we do not consider it necessary to include a section on the subject in the introduction.

Reviewer #2: The present work is very thorough and detailed.

I would like to ask the author why they removed the hospitalized elderly in the exclusion criteria, considering that most covid deaths occurred in hospital facilities. Modify the conclusions to strengthen the sense of the review.

In agreement with the reviewer's analysis, the group also understands that hospitalized elderly people should not be considered as exclusion criteria. Criteria modified in the text.

Reviewer #3: The authors present a systematic review protocol with the objective of evaluating factors associated with higher mortality from Covid-19 in the elderly.

Although the authors state in the introduction "there is still no systematic and comprehensive evidence worldwide on such issues specifically with elderly people" (lines 113 and 114), a quick search finds systematic reviews with meta-analysis with the same objective already published (doi: 10.1177/23337214211057392./doi: 10.3390/ijerph18158008/ https://doi.org/10.1556/2060.2022.00206) and the authors need to point out in the protocol whether what they propose may indicate different results from those published.

Finally, the protocol needs to consider the possibility of meta-regression in the methodology as it is a systematic review of observational studies, may have a shorter and more objective introduction and needs to be reviewed by a native English speaker to correct grammatical errors. As a result, there is no quality or originality for the protocol to be published.

The studies pointed out by the reviewer are relevant publications in the field and appear to be related to the topic agreed in this protocol.

However, our review aims to cover some of the methodological limitations presented in the articles cited and other studies carried out, such as: limitation of research time, restricted number of databases, restriction to the investigation of local data (one or a few countries) and research limited to biological aspects.

In this sense, we believe that the review proposed in our protocol is unprecedented in that it expands the investigation of mortality to include clinical (biological) and behavioral (non-biological) aspects, without restricting the location or language, with an extended time frame (2020-2023), considering classic epidemiological designs and with searches in an expanded number of databases.

Metaregression was considered in the topic of data analysis.

The article was revised again to adjust grammatical errors.

Respectfully,

The authors.

---

## [Decision Letter · Decision Letter 1]

17 Jan 2024

PONE-D-23-20685R1Factors associated with mortality of elderly people due to Covid-19: Protocol for systematic review and meta-analysisPLOS ONE

Dear Dr. Piuvezam,

Thank you for submitting your manuscript to PLOS ONE. After careful consideration, we feel that it has merit but does not fully meet PLOS ONE’s publication criteria as it currently stands. Therefore, we invite you to submit a revised version of the manuscript that addresses the points raised during the review process.

We look forward to receiving your revised manuscript.

Kind regards,

Rocco Franco

Academic Editor

PLOS ONE

Journal Requirements:

Additional Editor Comments:

Dear Authors,

please follow the reviewer's recommendation

Regards

Reviewers' comments:

Reviewer's Responses to Questions

**Comments to the Author**

1. Does the manuscript provide a valid rationale for the proposed study, with clearly identified and justified research questions?

Reviewer #2: Yes

Reviewer #3: Yes

Reviewer #4: Yes

Reviewer #5: Yes

2. Is the protocol technically sound and planned in a manner that will lead to a meaningful outcome and allow testing the stated hypotheses?

Reviewer #2: Yes

Reviewer #3: Yes

Reviewer #4: Yes

Reviewer #5: Yes

3. Is the methodology feasible and described in sufficient detail to allow the work to be replicable?

Reviewer #2: Yes

Reviewer #3: Yes

Reviewer #4: Yes

Reviewer #5: Yes

4. Have the authors described where all data underlying the findings will be made available when the study is complete?

Reviewer #2: Yes

Reviewer #3: Yes

Reviewer #4: Yes

Reviewer #5: Yes

5. Is the manuscript presented in an intelligible fashion and written in standard English?

Reviewer #2: Yes

Reviewer #3: Yes

Reviewer #4: Yes

Reviewer #5: Yes

6. Review Comments to the Author

You may also provide optional suggestions and comments to authors that they might find helpful in planning their study.

Reviewer #2: Dears colleagues

The paper has well-designed research methods, appropriate statistical analysis and a relatively good interpretation of the results.

-Please be sure to use only keywords accordingly to medical subject headings (Mesh word) for a better indexing.

.

I suggest you implement the abstract in order to make it more understandable to authors.

The introduction should be expanded perhaps by adding a section on covid diseases.

The conclusion is in accordance with the objectives of the research, its results and their interpretation, as well as the relevant literature.

Reviewer #3: The authors responded satisfactorily to the reviewers' recommendations and managed to improve the justification for yet another protocol on the topic in the introduction, including what it adds to what has already been published in other reviews. However, there are still corrections to be made.

In the introduction, after the phrase "Despite the wealth of studies produced and published", authors must cite the studies (doi: 10.1177/23337214211057392./doi:10.3390/ijerph18158008/ https://doi.org/10.1556/2060.2022.00206).

In the methodology, the text on meta-regression was inserted in Portuguese "Além disso, uma metaregressão poderá complementar a exploração das fontes da heterogeneidade.", and the main reason for considering metaregression in this protocol is not to explore sources of heterogeneity, but rather to allow the analysis of confounding factors associated with mortality in the elderly, allowing us to identify which are truly associated with COVID-19. They must correct and improve the text in English!

Reviewer #4: Comments to authors

Thank you for the opportunity to review (R1) of the manuscript titled “Factors associated with mortality of elderly people due to COVID-19: Protocol for systematic review and meta-analysis.”

Overall, the findings of this worldwide comprehensive study will benefit in planning and making targeted interventions on elderly people to reduce mortality due to COVID-19. Moreover countries will minimize the influences of pandemics / health crises on elder population.

I believe there are chances to more strengthen this manuscript. Here are the comments to be addressed by the authors.

1. Title

-Please write the word covid-19 in capital letters, and assure that it is consistent through the manuscript

2. Abstract

- Well written.

-Please spesify the version of the software will be used in this meta analysis?

3. Introduction

- Well written.

4. Methods and Methods

-In the exclusion criteria what is the reason in which the ahuthors will exclude Cross-sectional observational studies? Remember that currently cross sectional studies that can assess the prevalence and associated factors of mortality among elderly people due to COVID-19 are available. Please justify?

- In the data synthesis and analysis section, the stetment “ A quantitative synthesis of the results will be performed if the included studies are sufficiently homogeneous (75 to 100%) is not clear.

Please clarify?

-What is the statement “Além disso, uma metaregressão poderá complementar a exploração das fontes da heterogeneidade”? If it is meaningful please write in English or remove it.

-Please specify the methods of assessing the source of heterogeneity in this meta analysis. Subgroup analyses or meta regression?

- The author should specify how publcation bias will be assessed?

-Also the author should propose to see the small study effects or sensitivity analysis?

5. Discussion

- Well written.

6. Conclusion

- Well written.

7. Bibliographic references

- Well written.

Reviewer #5: The article is well written. The authors have taken the reviewers' comments into account. My additions to the article are:

Introduction

End the introduction with a sentence specifying the objective of the systematic review.

Methods

1. Evaluating patient mortality covid-19, in data extraction, sars- cov 2 variants also seems relevant during extraction

2. In the data analysis section, there is a sentence in Portuguese that needs to be translated into English “Além disso, uma metaregressão poderá complementar a exploração das fontes da heterogeneidade “ .

3. Specify the level of heterogeneity (low, moderate and high) with the I2 values and what you will do if the heterogeneity is high, e.g. sensitivity or subgroup analysis or metaregression.

4. Add a section on publication bias, e.g. Egger's test or Begg's funnel plot.

5. Add a sentence on ethics statement, although ethics committee approval is not required for meta-analysis.

Discussion

Add strenghs and limitations of the sudy, Implications of the results you find

7. PLOS authors have the option to publish the peer review history of their article (what does this mean?). If published, this will include your full peer review and any attached files.

Reviewer #2: **Yes: **Alessio Rosa

Reviewer #3: **Yes: **Ricardo Ney Cobucci

Reviewer #4: **Yes: **Aragaw Asfaw Hasen

Reviewer #5: **Yes: **Ben Bepouka'

---

## [Author Response · Author response to Decision Letter 1]

23 Feb 2024

Feb 23th, 2024

Rebuttal Letter for PLOS ONE

Thank you for considering our manuscript for publication at PLOS ONE.We are confident in PLOS ONE's reputation and dedication to scientific excellence, and hope that the revisions we have made to our manuscript are in line with the required publication standards. We believe that the suggestions contribute significantly to the quality and clarity of our article. 

The responses to each of the suggestions are detailed at the end of this message. 

Once again, we thank you for your consideration of our work and we are at your disposal for any further clarification that may be required. 

Sincerely,

Reviewer #2: Dears colleagues

The paper has well-designed research methods, appropriate statistical analysis and a relatively good interpretation of the results.

Please be sure to use only keywords accordingly to medical subject headings (Mesh word) for a better indexing.

I suggest you implement the abstract in order to make it more understandable to authors.

The introduction should be expanded perhaps by adding a section on covid diseases.

The conclusion is in accordance with the objectives of the research, its results and their interpretation, as well as the relevant literature.

All the terms entered in the search key have been reviewed and they are MESH as descriptors or between terms.

The summary has been improved.

As suggested, a paragraph has been added to the introduction about the COVID-19 disease.

Reviewer #3: The authors responded satisfactorily to the reviewers' recommendations and managed to improve the justification for yet another protocol on the topic in the introduction, including what it adds to what has already been published in other reviews. However, there are still corrections to be made.

In the introduction, after the phrase "Despite the wealth of studies produced and published", authors must cite the studies (doi: 10.1177/23337214211057392./doi:10.3390/ijerph18158008/ https://doi.org/10.1556/2060.2022.00206).

In the methodology, the text on meta-regression was inserted in Portuguese "Além disso, uma metaregressão poderá complementar a exploração das fontes da heterogeneidade.", and the main reason for considering metaregression in this protocol is not to explore sources of heterogeneity, but rather to allow the analysis of confounding factors associated with mortality in the elderly, allowing us to identify which are truly associated with COVID-19. They must correct and improve the text in English!

We have included the citation to the studies after the sentence, as suggested.

We have revised the meta-regression sentence and we enhanced the English text.

Reviewer #4: Comments to authors

Thank you for the opportunity to review (R1) of the manuscript titled “Factors associated with mortality of elderly people due to COVID-19: Protocol for systematic review and meta-analysis.”

Overall, the findings of this worldwide comprehensive study will benefit in planning and making targeted interventions on elderly people to reduce mortality due to COVID-19. Moreover countries will minimize the influences of pandemics / health crises on elder population.

I believe there are chances to more strengthen this manuscript. Here are the comments to be addressed by the authors.

1. Title

-Please write the word covid-19 in capital letters, and assure that it is consistent through the manuscript

2. Abstract

- Well written.

-Please specify the version of the software will be used in this meta analysis?

3. Introduction

- Well written.

4. Methods and Methods

-In the exclusion criteria what is the reason in which the ahuthors will exclude Cross-sectional observational studies? Remember that currently cross sectional studies that can assess the prevalence and associated factors of mortality among elderly people due to COVID-19 are available. Please justify?

- In the data synthesis and analysis section, the stetment “ A quantitative synthesis of the results will be performed if the included studies are sufficiently homogeneous (75 to 100%) is not clear.

Please clarify?

-What is the statement “Além disso, uma metaregressão poderá complementar a exploração das fontes da heterogeneidade”? If it is meaningful please write in English or remove it.

-Please specify the methods of assessing the source of heterogeneity in this meta analysis. Subgroup analyses or meta regression?

- The author should specify how publcation bias will be assessed?

-Also the author should propose to see the small study effects or sensitivity analysis?

5. Discussion

- Well written.

6. Conclusion

- Well written.

7. Bibliographic references

- Well written.

We have capitalized the term COVID-19 consistently throughout the manuscript, as per the suggestion.

With regard to the criterion for excluding cross-sectional studies, we believe that these studies collect data at a single point in time and therefore have limitations in terms of causal inference and temporal follow-up. Consequently, we chose to take advantage of the benefits offered by longitudinal studies in this context. This becomes particularly advantageous when considering that certain COVID-19 patients have undergone significant follow-up, which may facilitate the identification of additional factors associated with mortality from the disease. In addition, we postulate that longitudinal studies are able to provide associated factors, aligning with the objectives of our protocol. Specifically, cohort studies not only provide associated factors, but also delve into risk factors. Given the possibility of data homogeneity, this increases our ability to carry out a more robust meta-analysis.

Regarding the quantitative synthesis section, we would like to inform you that a meta-analysis will be carried out if the studies show sufficient homogeneity. We have revised the sentence in the manuscript.

We have revised the meta-regression sentence and enhanced the English text.

The funnel plot and/or sensitivity analysis will be used to assess publication bias. Added to the study, according to the reviewer's observation.

Reviewer #5: The article is well written. The authors have taken the reviewers' comments into account. My additions to the article are:

Introduction

End the introduction with a sentence specifying the objective of the systematic review.

Methods

1. Evaluating patient mortality covid-19, in data extraction, sars- cov 2 variants also seems relevant during extraction

2. In the data analysis section, there is a sentence in Portuguese that needs to be translated into English “Além disso, uma metaregressão poderá complementar a exploração das fontes da heterogeneidade “ .

3. Specify the level of heterogeneity (low, moderate and high) with the I2 values and what you will do if the heterogeneity is high, e.g. sensitivity or subgroup analysis or metaregression.

4. Add a section on publication bias, e.g. Egger's test or Begg's funnel plot.

5. Add a sentence on ethics statement, although ethics committee approval is not required for meta-analysis.

Discussion

Add strenghs and limitations of the sudy, Implications of the results you find

We have added the sentence with the objective of the review to the end of the introduction.

We have added the types of variants studied to the data extraction.

We have revised the meta-regression sentence and enhanced the English text.

We have added to the text the values to be considered in the heterogeneity with I²: near 25% will be considered low, near 50%, moderate, and near 75%, high.

We have revised the meta-regression sentence and enhanced the English text.

The funnel plot and/or sensitivity analysis will be used to assess publication bias. Added to the study, according to the reviewer's observation.

A section on ethics statements was included.

We have added a section on the strengths, limitations and implications of the study at the end of the discussion, as suggested.

Respectfully,

The authors.

---

## [Decision Letter · Decision Letter 2]

2 Apr 2024

Factors associated with mortality of elderly people due to Covid-19: Protocol for systematic review and meta-analysis

PONE-D-23-20685R2

Dear Dr. Piuvezam,

We’re pleased to inform you that your manuscript has been judged scientifically suitable for publication and will be formally accepted for publication once it meets all outstanding technical requirements.

Kind regards,

Rocco Franco

Academic Editor

PLOS ONE

Additional Editor Comments (optional):

Congratulations, based also on the reviewer's comments, this paper can be accepted.

Regards

Reviewers' comments:

Reviewer's Responses to Questions

**Comments to the Author**

1. Does the manuscript provide a valid rationale for the proposed study, with clearly identified and justified research questions?

Reviewer #3: Yes

Reviewer #5: Yes

2. Is the protocol technically sound and planned in a manner that will lead to a meaningful outcome and allow testing the stated hypotheses?

Reviewer #3: Yes

Reviewer #5: Yes

3. Is the methodology feasible and described in sufficient detail to allow the work to be replicable?

Reviewer #3: Yes

Reviewer #5: Yes

4. Have the authors described where all data underlying the findings will be made available when the study is complete?

Reviewer #3: Yes

Reviewer #5: Yes

5. Is the manuscript presented in an intelligible fashion and written in standard English?

Reviewer #3: Yes

Reviewer #5: Yes

6. Review Comments to the Author

You may also provide optional suggestions and comments to authors that they might find helpful in planning their study.

Reviewer #3: I have no more comments. My comments for the last version of this manuscript have been addressed. Congratulations!

Reviewer #5: the manuscript is well written . the authors found that there is a notable absence of systematic review studies

consolidating knowledge about both biological and non-biological factors among

elderly individuals who succumbed to SARS-CoV-2 infection, considering global data.

Given the consensus that the elderly population is the most vulnerable demographic

to serious outcomes and deaths during pandemics, addressing these gaps is crucial

for the informed development of public policies, enabling countries to minimize the

impacts on this population, particularly during health crises such as the COVID-19

pandemic.

And in this context, we found that this systematic review could contribute to the science.

7. PLOS authors have the option to publish the peer review history of their article (what does this mean?). If published, this will include your full peer review and any attached files.

Reviewer #3: **Yes: **Ricardo Ney Cobucci

Reviewer #5: **Yes: **Ben Bepouka

---

## [Editor Report · Acceptance letter]

8 Apr 2024

PONE-D-23-20685R2 

PLOS ONE

Dear Dr. Piuvezam, 

I'm pleased to inform you that your manuscript has been deemed suitable for publication in PLOS ONE. Congratulations! Your manuscript is now being handed over to our production team.

Kind regards, 

on behalf of

Dr. Rocco Franco 

Academic Editor

PLOS ONE